# α-synuclein strains that cause distinct pathologies differentially inhibit proteasome

Genjiro Suzuki[1]*, Sei Imura[1,2], Masato Hosokawa[1], Ryu Katsumata[1], Takashi Nonaka[1], Shin-Ichi Hisanaga[2], Yasushi Saeki[3], Masato Hasegawa[1]*

[1]Dementia Research Project, Tokyo Metropolitan Institute of Medical Science, Tokyo, Japan; [2]Laboratory of Molecular Neuroscience, Department of Biological Sciences, Tokyo Metropolitan University, Tokyo, Japan; [3]Protein Metabolism Project, Tokyo Metropolitan Institute of Medical Science, Tokyo, Japan

**Abstract** Abnormal α-synuclein aggregation has been implicated in several diseases and is known to spread in a prion-like manner. There is a relationship between protein aggregate structure (strain) and clinical phenotype in prion diseases, however, whether differences in the strains of α-synuclein aggregates account for the different pathologies remained unclear. Here, we generated two types of α-synuclein fibrils from identical monomer and investigated their seeding and propagation ability in mice and primary-cultured neurons. One α-synuclein fibril induced marked accumulation of phosphorylated α-synuclein and ubiquitinated protein aggregates, while the other did not, indicating the formation of α-synuclein two strains. Notably, the former α-synuclein strain inhibited proteasome activity and co-precipitated with 26S proteasome complex. Further examination indicated that structural differences in the C-terminal region of α-synuclein strains lead to different effects on proteasome activity. These results provide a possible molecular mechanism to account for the different pathologies induced by different α-synuclein strains.

*For correspondence:
suzuki-gj@igakuken.or.jp (GS);
hasegawa-ms@igakuken.or.jp
(MH)

Competing interests: The authors declare that no competing interests exist.

## Introduction

Misfolding and aggregation of normally soluble proteins are common pathological features of many neurodegenerative diseases, including Alzheimer's, Parkinson's, Creutzfeldt–Jacob and Huntington's diseases (*Ross and Poirier, 2004*). For example, Parkinson's disease (PD), dementia with Lewy bodies (DLB) and multiple system atrophy (MSA) are characterized by accumulation of misfolded α-synuclein in neuronal and/or glial cells, and therefore these diseases are termed α-synucleinopathies. In PD and DLB, α-synuclein pathologies are mainly observed in neurons in the form of Lewy bodies (LBs) and Lewy neurites (LNs) (*Baba et al., 1998*), while glial cytoplasmic inclusions (GCIs) are seen in oligodendrocytes in MSA (*Wakabayashi et al., 1998*). The abnormal α-synuclein observed in brains of patients is accumulated as fibrous or filamentous forms with cross-β structures (*Araki et al., 2019*; *Spillantini et al., 1997*), existing in phosphorylated and partially ubiquitinated states (*Fujiwara et al., 2002*; *Hasegawa et al., 2002*). These abnormal α-synuclein species exhibit seeding activity for prion-like conversion, being similar in this respect to the infectious forms of prion protein (PrP) causing Creutzfeldt-Jakob disease (CJD) and bovine spongiform encephalopathy (*Goedert, 2015*). Various other neurodegenerative disease-related proteins, including amyloid-β, tau and TDP-43, can also propagate through neural networks in a similar manner.

α-Synuclein is a natively unfolded protein of 140 amino acid residues, normally found in both soluble and membrane-associated fractions and localized in synaptic termini. Although its physiological function has not been fully clarified, it appears to be involved in the regulation of SNARE complex and in dopamine production. Disease-linked missense mutations and multiplication of the *SNCA*

gene encoding α-synuclein have been reported in familial forms of α-synucleinopathies, indicating that structural changes and overexpression of α-synuclein protein are involved in the development of α-synucleinopathies (*Wong and Krainc, 2017*).

Recombinant soluble α-synuclein proteins purified from bacterial cells form amyloid-like fibrils that are morphologically and physicochemically similar to those observed in patients' brains (*Araki et al., 2019*; *Goedert, 2015*). These synthetic α-synuclein fibrils can act as seeds and induce seeded aggregation of α-synuclein in cultured cells or primary cultured neurons, as well as in animal brains. Intracerebral inoculation of synthetic α-synuclein fibrils induces phosphorylated and ubiquitinated α-synuclein pathologies even in wild-type (WT) mice (*Luk et al., 2012*; *Masuda-Suzukake et al., 2013*). It has also been reported that extracts from brains of patients with α-synucleinopathies induce α-synuclein pathologies in cellular and animal models (*Bernis et al., 2015*; *Watts et al., 2013*). In addition, recent studies have suggested that α-synuclein strains with distinct conformations exist, which is a characteristic of prions (*Bousset et al., 2013*; *Gribaudo et al., 2019*; *Guerrero-Ferreira et al., 2019*; *Peelaerts and Baekelandt, 2016*; *Peelaerts et al., 2015*; *Shahnawaz et al., 2020*; *Woerman et al., 2019*). Synthetic α-synuclein fibrils formed under different physiological conditions in vitro show distinct seeding activities and cytotoxicity in cultured cells and rat brains. Furthermore, MSA brain extracts exhibit distinct infectivity compared to PD or control brain extracts in cultured cells or mice expressing mutant A53T or WT α-synuclein (*Lau et al., 2020*; *Peng et al., 2018*; *Prusiner et al., 2015*; *Woerman et al., 2019*; *Woerman et al., 2015*).

These observations support the idea that α-synuclein shows prion-like behavior, because they can be accounted for by a typical hallmark of the prion phenomenon, that is, the presence of strains. In prion diseases, the variety of strains that can be differentiated in terms of the clinical signs, incubation period after inoculation, and the vacuolation lesion profiles in the brain of affected animals is due to structural differences of PrP aggregates, as identified by biochemical analyses including glycosylation profile, electrophoretic mobility, protease resistance, and sedimentation. These PrP strains are thought to correspond to different conformations of PrP aggregates, as demonstrated for the yeast prion [*PSI*$^+$], which induces aggregates of Sup35p (*Ohhashi et al., 2010*). Thus, as the case of prion disease, differences of lesions among α-synucleinopathies are thought to be caused by conformational heterology of α-synuclein assemblies, probably amyloid-like fibrils (*Lau et al., 2020*; *Schweighauser et al., 2020*; *Shahnawaz et al., 2020*). However, little is known about how conformational differences of protein aggregates induce a variety of lesions, not only in prion disease, but also in other neurodegenerative diseases, such as α-synucleinopathies.

In this study, we prepared two α-synuclein assemblies from identical wild-type α-synuclein monomer under different conditions, and established that they have distinct conformations, that is, we succeeded to generating two α-synuclein fibrils from the same monomer. We examined their seeding abilities to convert endogenous soluble α-synuclein monomers into phosphorylated aggregates in mice and primary-cultured neurons, indicating the formation of two α-synuclein strains. Notably, only one strain induced the accumulation of ubiquitinated proteins, as well as phosphorylated α-synuclein aggregates, indicating the ability of this strain to inhibit proteasome activity. Moreover, only this strain strongly inhibited proteasome activity and co-precipitated with purified 26S proteasome complex in vitro. Structural studies suggested that the C-terminal region plays a key role in the different properties of the two strains. Taken together, these results provide a possible molecular mechanism to account for the different lesions induced by distinct α-synuclein strains.

## Results

### Formation of amyloid-like fibrils from α-synuclein proteins in vitro

We generated two distinct α-synuclein assemblies from the identical purified α-synuclein monomer using the method described in the previous report, with minor modifications (*Figure 1A*; *Bousset et al., 2013*; *Lau et al., 2020*). Specifically, we prepared recombinant α-synuclein monomer and agitated it in the presence or absence of salt at a physiological concentration. In the presence of physiological salt (30 mM Tris, pH 7.5, 150 mM KCl), the monomer formed a cloudy solution of assemblies with higher turbidity, while in the absence of salt (30 mM Tris, pH 7.5), a clear solution containing assemblies with lower turbidity was formed (*Figure 1A and B*). Almost all of the α-synuclein was present as aggregates rather than soluble oligomers under both conditions (*Figure 1—*

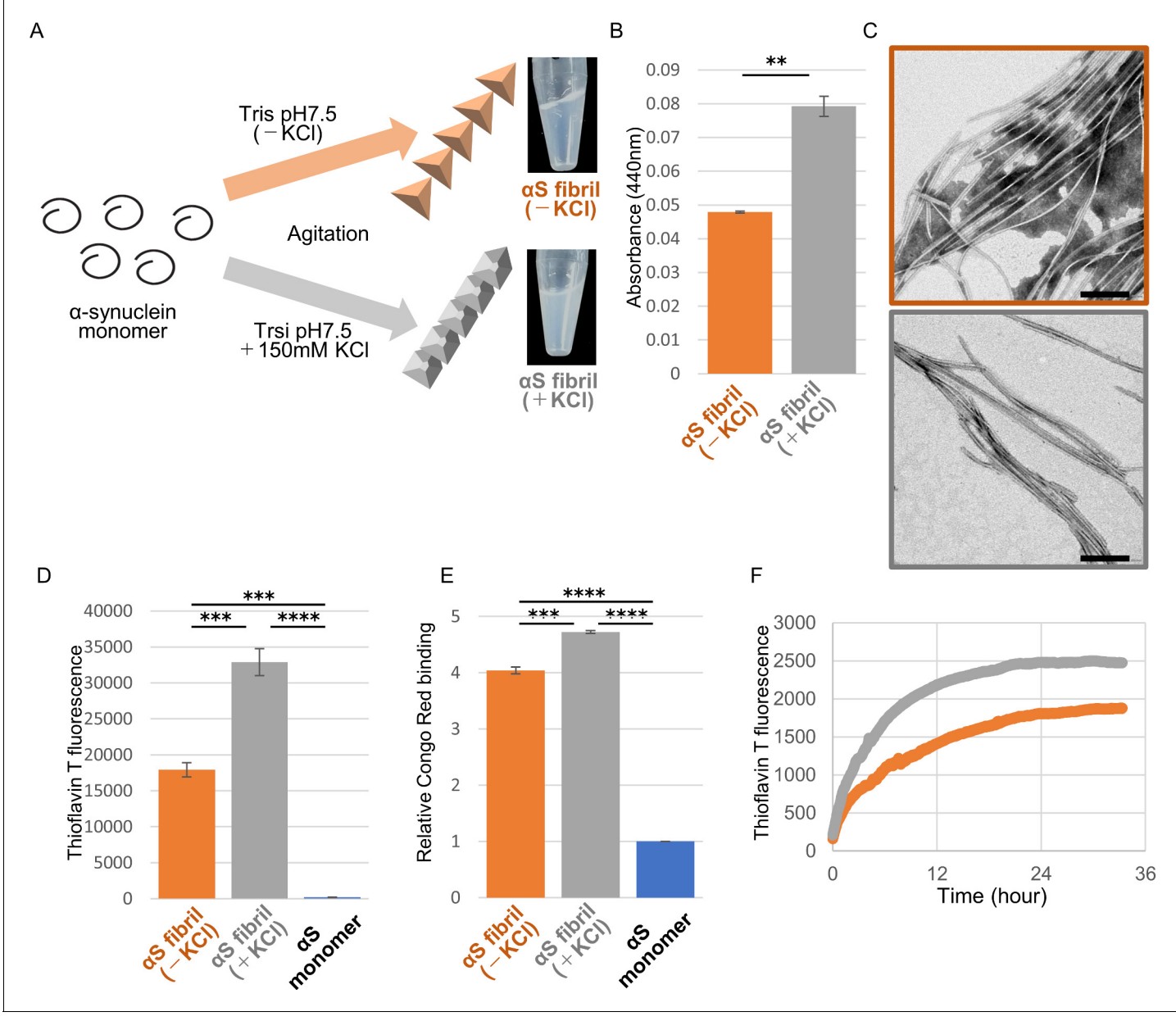

**Figure 1.** Preparation and characterization of two α-synuclein strains. (**A**) Schematic representation of two α-synuclein (αS) strains (left) and the resulted two α-synuclein assemblies (right). (**B**) Turbidity of these assemblies. Analysis was performed using student t test. (mean ± S.E.M; n = 3) Formation of aggregates rather than soluble oligomer is shown in *Figure 1—figure supplement 1*. (**C**) Transfer electron microscopy (TEM) images of these assemblies. Scale bar, 200 nm. (**D**) Thioflavin T fluorescence of these assemblies and αS monomer. Analysis was performed using one-way ANOVA and Tukey post hoc test. (mean ± S.E.M; n = 3) (**E**) Congo red bindings of these strains and αS monomer. (mean ± S.E.M; n = 3) (**F**) In vitro seeding activity of these strains. α-synuclein monomers were incubated with α-synuclein fibril (-) (orange) or α-synuclein fibril (+) (gray). Kinetics of Thioflavin T fluorescence were shown. Analysis was performed using one-way ANOVA and Tukey post hoc test. **p<0.01, ***p<0.001, ****p<0.0001.
The online version of this article includes the following source data and figure supplement(s) for figure 1:

**Source data 1.** Quantification for graph in *Figure 1B,D and E*.
**Source data 2.** Three independent data in *Figure 1F*.
**Figure supplement 1.** Formation of α-synuclein aggregates.

*figure supplement 1*; *Thibaudeau et al., 2018*). Both assemblies showed fibrillar morphology, but the previously reported ribbon-like morphology was not observed (*Figure 1C*; *Bousset et al., 2013*). Therefore, we will refer to the former assemblies as α-synuclein fibrils (+) and the later assemblies as α-synuclein fibrils (-). Both fibrils were stained with Thioflavin T and Congo red, indicating their amyloid-like nature (*Figure 1D and E*). The differences of Thioflavin T fluorescence and Congo red binding of these α-synuclein fibrils suggested that these fibrils have a different structures each other. Next, we examined the in vitro seeding activity of these fibrils and found that α-synuclein fibrils (+) showed higher seeding activity than α-synuclein fibrils (-) in vitro (*Figure 1F*). These results indicated that we had successfully prepared two distinct types of α-synuclein fibrils from the same monomer.

## Formation of phosphorylated α-synuclein pathology by injection of α-synuclein strains into mouse brain

We next investigated whether there was a strain-dependent difference of prion-like propagation in mouse brain. We injected α-synuclein fibrils (+) and α-synuclein fibrils (-) into striatum of wild-type mice, and after one month, we examined the accumulation of phosphorylated α-synuclein deposits resembling those observed in patients' brains (*Luk et al., 2012*; *Masuda-Suzukake et al., 2013*). In contrast to the in vitro seeding activities, α-synuclein fibrils (-) induced Lewy body/Lewy neurite-like abnormal phosphorylated α-synuclein deposits through the mouse brain, including cortex, striatum and corpus callosum, whereas few phosphorylated α-synuclein deposits were induced by α-synuclein fibrils (+) (*Figure 2A* and *Figure 2—figure supplement 1*). We used three mice each for this experiment and confirmed the reproducibility (*Figure 2—figure supplement 2*). The deposits induced by α-synuclein fibrils (-) were also positive for ubiquitin staining in cortex, like Lewy bodies and Lewy neurites (*Figure 2B*). Compared with phosphorylated α-synuclein, which is the most representative marker in the α-synucleinopathies, ubiquitin staining is weaker, thus we could find only few ubiquitin positive deposits in striatum and corpus callosum (data not shown). Taken together, there was a strain-dependent difference of α-synuclein fibril formation from the identical α-synuclein monomer when the two strains were inoculated into the mouse brain, and α-synuclein fibrils (-) had higher prion-like seeding activity than α-synuclein fibrils (+) in mouse brain, contrary to the in vitro seeding results (*Figure 1F*).

## Formation of phosphorylated α-synuclein pathology induced by the two α-synuclein strains in cultured neurons

To further study the difference in the formation of pathological α-synuclein aggregates in neurons, we compared the ability of the two α-synuclein strains to induce seed-dependent aggregation of α-synuclein in primary mouse cortical cells. When primary mouse cortical cells from non-transgenic mice were treated with α-synuclein fibrils for two weeks, we observed a dramatic increase of phosphorylated α-synuclein accumulation, which is also positive for ubiquitin, only in the case of α-synuclein fibrils (-), while little accumulation was seen with α-synuclein fibrils (+) (*Figure 3A*). These results are consistent with those observed in the mouse brain. To investigate whether these accumulations occur in neurons, we performed double staining for phosphorylated α-synuclein and neuronal markers or an astrocyte marker using cells treated with α-synuclein fibrils (-) (*Figure 3—figure supplement 1*). We found that phosphorylated α-synuclein accumulations were seen in neurons, not in astrocytes. Thus, α-synuclein fibrils (-) could induce phosphorylated α-synuclein accumulations in primary mouse cortical neurons.

Next, we performed biochemical analysis of detergent-insoluble α-synuclein prepared from these cells. Phosphorylated α-synuclein were accumulated in cells treated with both α-synuclein fibrils (-) and α-synuclein fibrils (+) (*Figure 3B and C*). However, α-synuclein fibrils (-) induced a greater accumulation of phosphorylated α-synuclein than did α-synuclein fibrils (+) (*Figure 3D*). The phosphorylated and aggregated α-synuclein in these cells was found to be endogenous mouse α-synuclein, indicating that the introduced human α-synuclein fibrils worked as seeds (*Figure 3B*). We also examined the accumulation of detergent-insoluble ubiquitinated proteins and found that not only ubiquitinated α-synuclein, but also other ubiquitinated proteins were accumulated in cells treated with α-synuclein fibrils (-). There was no significant increase of ubiquitinated protein accumulation in cells treated with α-synuclein fibrils (+) (*Figure 3B and D*). These results indicated that only α-synuclein

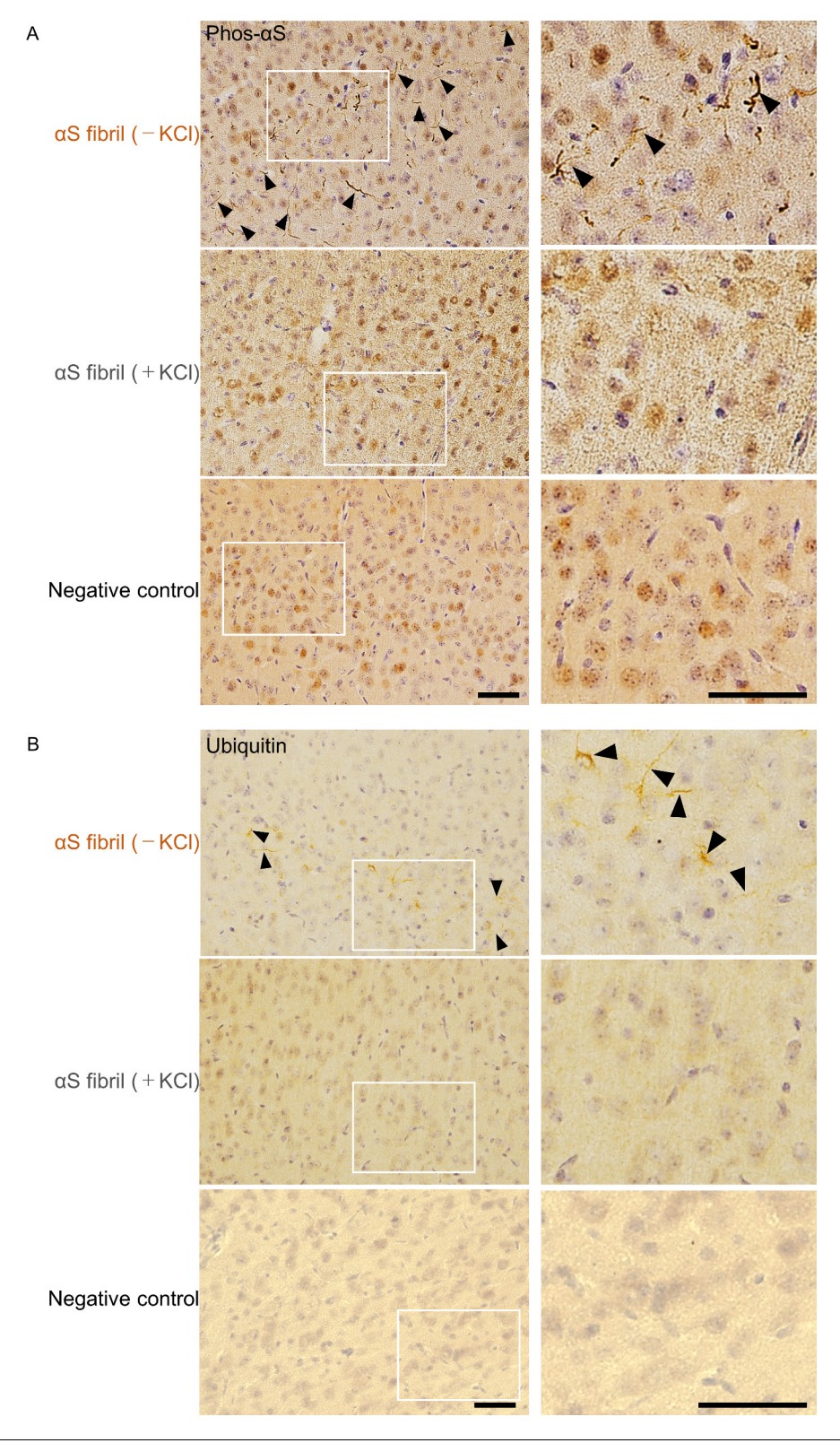

**Figure 2.** Comparison of pathologies in WT mouse brains by inoculation of two α-synuclein strains. (**A**) Distribution of phosphorylated α-synuclein pathology in mouse brain. α-synuclein fibrils (-) or α-synuclein fibrils (+) were injected into the striatum of WT mouse brain and stained with phosphorylated α-synuclein antibody 1 month after injection. Regions surrounded by white rectangles in left panels are magnified and shown in right panels. Typical pathological α-synuclein deposits are indicated by the arrowheads. Sections were counterstained with hematoxylin. Representative

*Figure 2 continued on next page*

*Figure 2 continued*

images of cortex of mice injected with α-synuclein fibrils (-) (upper), α-synuclein fibrils (+) (middle) or not injected mice as negative controls (lower) are shown. Representative images of corpus callosum and striatum are shown in *Figure 2—figure supplement 1*. Representative images of cortex of other two mice injected with α-synuclein fibrils are also shown in *Figure 2—figure supplement 2*. (B) Distribution of ubiquitin pathology in mouse brain. α-synuclein fibrils (-) or α-synuclein fibrils (+) were injected into the striatum of WT mouse brain and stained with ubiquitin antibody 1 month after injection. Representative images of cortex of mice injected with α-synuclein fibrils (-) (upper), α-synuclein fibrils (+) (middle) or not injected mice as negative controls (lower) are shown. Regions surrounded by white rectangles in left panels are magnified and shown in right panels. Typical pathological ubiquitin positive deposits are indicated by the arrowheads. Sections were counterstained with hematoxylin. Scale bars, 50 μm.

The online version of this article includes the following figure supplement(s) for figure 2:

**Figure supplement 1.** Comparison of pathologies in WT mouse brains by inoculation of two α-synuclein strains.

**Figure supplement 2.** Images of cortex of other two mice injected with the α-synuclein fibrils.

fibrils (-) induced much accumulation of phosphorylated α-synuclein and ubiquitinated proteins in primary mouse cortical neurons, in accordance with the findings in mouse brain. Seed dependent aggregation of α-synuclein must depend on the efficiency of fibril uptake by cells. We confirmed whether the two α-synuclein fibrils were taken up into primary mouse cortical cells with similar efficiency. Cells were treated with the two α-synuclein fibrils or buffer. A day after treatment, we treated cells with trypsin for digestion of extracellular fibrils, collected them and examined the α-synuclein fibrils taken up into the cells by western blotting of sarkosyl insoluble fractions. We found no α-synuclein signal in pellet fraction of cells with buffer control indicating that detected α-synuclein signals were originated from added fibrils. We found α-synuclein fibrils (-), which induced much α-synuclein accumulation, were taken up into the cells lesser than α-synuclein fibrils (+), demonstrating that α-synuclein fibrils (-) efficiently caused the accumulation of α-synuclein and ubiquitinated proteins in spite of their inefficient uptake into the cells (*Figure 3—figure supplement 2*).

## Different interactions of α-Synuclein strains with 26S Proteasome

The above results motivated us to examine proteasome activity in the presence of these two types of α-synuclein fibrils. We purified 26S proteasome complex from budding yeast expressing FLAG-tagged Rpn11p, a subunit of 19S regulatory complex (*Figure 4—figure supplement 1*; *Saeki et al., 2005*). The activity of the purified 26S proteasome was examined in the presence or absence of α-synuclein fibrils. The eukaryotic proteasome has three active subunits, β1, β 2 and β 5, each displaying a specific catalytic activity, trypsin-like, chymotrypsin-like and caspase-like activity, respectively. Thus, we examined the chymotrypsin-like (*Figure 4A*), caspase-like (*Figure 4B*) and trypsin-like (*Figure 4C*) activities of 26S proteasome in the presence of these α-synuclein fibrils using each fluorogenic peptide substrate in vitro. The catalytic activity of 26S proteasome was drastically impaired in the presence of α-synuclein fibrils (-), whereas α-synuclein fibrils (+) were ineffective (*Figure 4A–C*). These results accord with our mouse and primary-cultured neuron data, i.e., only α-synuclein fibrils (-) could induce much accumulation of phosphorylated α-synuclein and ubiquitinated proteins. Co-aggregation of functional proteins with misfolded protein aggregates may cause the impairment of their function. Therefore, we investigated the interaction of 26S proteasome with the fibrils. α-Synuclein fibrils were mixed with purified 26S proteasome, and the mixture was centrifuged. The supernatants and precipitates were analyzed by western blotting. We found that both α-synuclein fibrils (-) and α-synuclein fibrils (+) were fractionated to the pellet fractions (*Figure 4D*). However, only α-synuclein fibrils (-) co-precipitated with 26S proteasome, while α-synuclein fibrils (+) did not (*Figure 4E*). Taken together, these results indicate that only α-synuclein fibrils (-) interact with 26S proteasome and impair the proteasome activity.

If α-synuclein fibrils (-) bind other proteins, such as 26S proteasome, while α-synuclein fibrils (+) do not, the structures of the two types of fibrils should be distinguishable. Indeed, we did observe slight differences between these two types of fibrils (*Figure 1*). Amyloid-like fibrils are composed of core regions consisting of β-sheet-rich rigid structure, and exposed regions, which might interact with other molecules. The core regions tend to be resistant to protease attack, whereas exposed regions tend to be easily digested by protease. To identify the core regions and exposed regions of these α-synuclein fibrils, we carried out limited proteolysis with proteinase K, followed by mass spectrometric analysis (*Figure 4F*; *Suzuki et al., 2012*). We found that α-synuclein fibrils (-) have a smaller core region (amino acid residues 39–96 and 40–94; m/z 5656 and 5268, respectively) corresponding

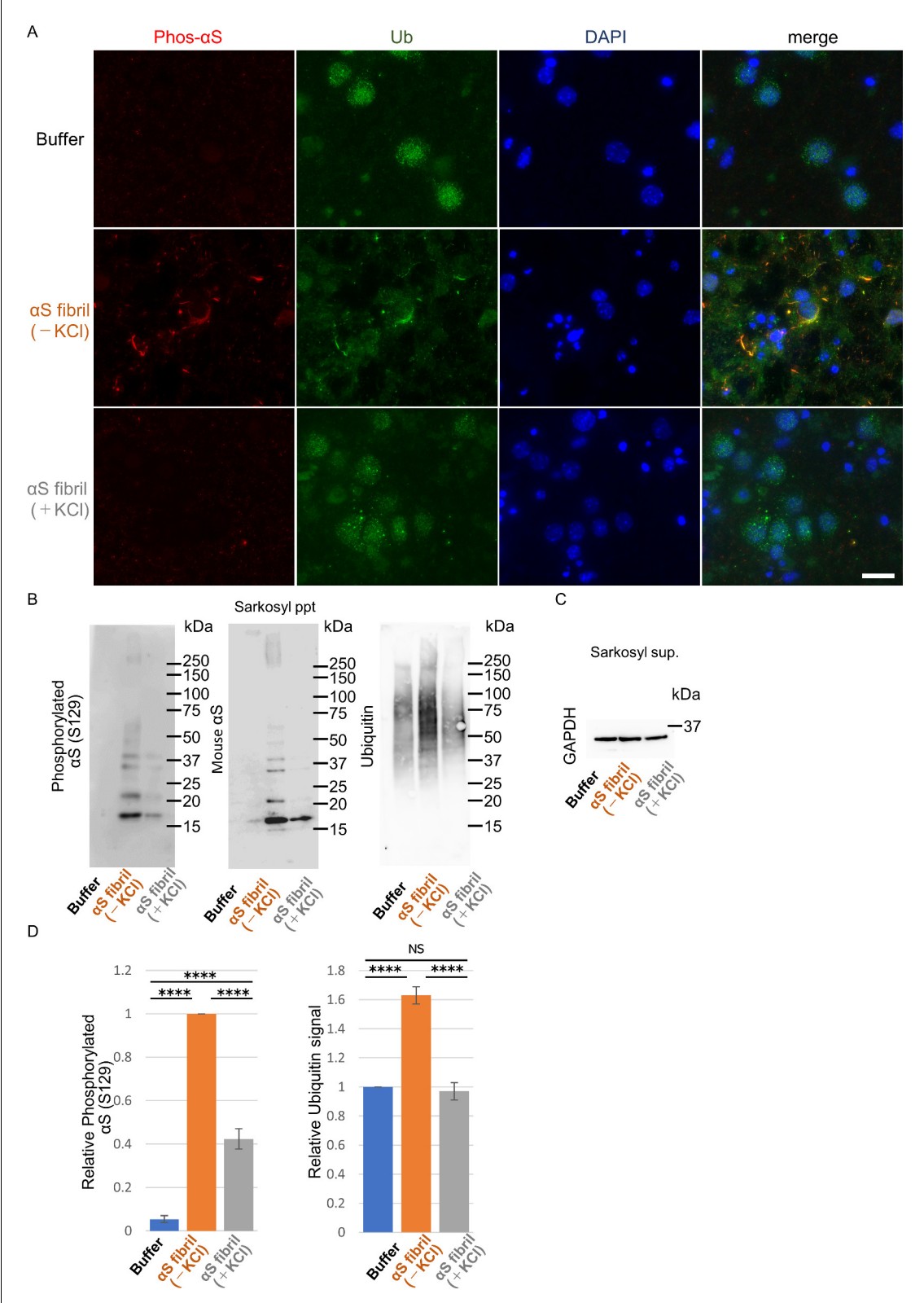

**Figure 3.** Seeding activities α-synuclein strains in primary mouse cortical neurons. The two α-synuclein fibrils were transduced into primary mouse cortical cells. 14 days after fibril transduction, the accumulation of abnormal phosphorylated α-synuclein and ubiquitinated aggregates are detected by immunofluorescence microscopy and western blotting. (**A**) Phosphorylated α-synuclein (Phos-αS), ubiquitin (Ub) and nuclei (DAPI) were stained. Scale bar, 20 μm. Images of cells double stained with antibody against phosphorylated α-synuclein and neuronal or astrocyte markers are shown in

*Figure 3 continued on next page*

*Figure 3 continued*

*Figure 3—figure supplement 1*. Uptake of these α-synuclein fibrils into cells are shown in *Figure 3—figure supplement 2*. (B) Detection of sarkosyl insoluble phosphorylated α-synuclein (left), endogenous mouse α-synuclein (center) and ubiquitinated proteins (right) by western blotting. (C) Detection of sarkosyl soluble GAPDH as a loading control. (D) The quantification data of sarkosyl insoluble phosphorylated α-synuclein (left) and ubiquitinated proteins (right) shown in (B) (mean ± S.E.M; n = 3). Analysis was performed using one-way ANOVA and Tukey post hoc test. ****p<0.0001.

The online version of this article includes the following source data and figure supplement(s) for figure 3:

**Source data 1.** Quantification for graph in *Figure 3D* and *Figure 3—figure supplement 2C*.
**Figure supplement 1.** Double stained Images of phosphorylated α-synuclein and neuronal or astrocyte markers.
**Figure supplement 2.** Uptake of the α-synuclein fibrils into the cells.

to the NAC region (amino acid residues 61–95), which had previously been reported as the core region of α-synuclein fibrils (*Guerrero-Ferreira et al., 2018*; *Li et al., 2018*). In contrast, α-synuclein fibrils (+) had a larger core region, extending to the C-terminal regions (residues 31–109, 28–118 and 25–132; m/z 7860, 9115 and 10907, respectively), indicating that α-synuclein fibrils (-) have amyloid structure with a more exposed C-terminal region than α-synuclein fibrils (+). To confirm these results, we performed dot-blot analysis using various antibodies against different regions of α-synuclein protein. Native α-synuclein fibrils were spotted on nitrocellulose membranes and detected by applying various α-synuclein antibodies. Antibodies raised against the N-terminal region of α-synuclein (residues 1–10) and the NAC region (91-99) almost equally recognized both types of α-synuclein fibrils, indicating that they have similar structure in the N-terminal and NAC regions (*Figure 4G*). However, as expected from the above data, the antibodies raised against C-terminal regions (amino acid 115–122 and 131–140) bound more strongly to α-synuclein fibrils (-) than to α-synuclein fibrils (+), supporting the idea that the C-terminal region of α-synuclein fibrils (-) is more exposed than that of α-synuclein fibrils (+). Thus, we considered that the C-terminal region of α-synuclein fibrils (-) might interact with 26S proteasome and impair its activities. Indeed, we previously reported that C-terminally truncated α-synuclein fibrils induced pathology in mouse brain less potently than did full-length α-synuclein fibrils, even though the C-terminally truncated α-synuclein fibrils had higher seeding activity in vitro (*Terada et al., 2018*). To confirm this, we next examined the seeding activity of C-terminally truncated fibrils formed in the absence of salt in primary-cultured neurons. We prepared C-terminally truncated α-synuclein monomer (residues 1–120) and agitated it in the absence of salt. The resulting assemblies showed fibrillar morphology and thioflavin T binding (data not shown), and we refer to them as αSΔC20 fibrils (-). We treated primary mouse cortical cells with αSΔC20 fibrils (-) and examined the accumulation of phosphorylated α-synuclein and ubiquitinated proteins. As expected, we observed little accumulation of these proteins (*Figure 5A and B*). Next, we examined the activities of 26S proteasome in the presence of αSΔC20 fibrils (-) as shown in *Figure 4A–C*. The catalytic activity of 26S proteasome was not impaired in the presence of αSΔC20 fibrils (-) (*Figure 5C*). We also investigated the interaction of 26S proteasome with αSΔC20 fibrils (-) as shown in *Figure 4D and E* and found that αSΔC20 fibrils (-) did not co-precipitated with 26S proteasome (*Figure 5D and E*). Considering all these results, we can conclude that the C-terminal region of α-synuclein is exposed only in α-synuclein fibrils (-) and this region interacts with 26S proteasome and impairs its activity.

## Discussion

According to the prion hypothesis, differences in disease symptoms and lesions are caused by differences in the conformation of strains. Therefore, if α-synuclein is prion-like, differences in the structure of α-synuclein aggregates should cause the differences in the lesions observed in various α-synucleopathies. Recent analyses have shown that introduction of extracts from brains of patients with PD and MSA into mice and cells induces different pathologies (*Prusiner et al., 2015*; *Tarutani et al., 2018*; *Woerman et al., 2019*; *Woerman et al., 2015*). Similar results were obtained with brain extracts of patients with tauopathies, in which tau is abnormally accumulated (*Narasimhan et al., 2017*; *Saito et al., 2019*). Moreover, it has been clarified that TDP-43 aggregates have a different structure depending on the disease, and different lesions appear when the patients' brain extracts are introduced into cells or animals (*Laferrière et al., 2019*; *Nonaka et al., 2013*; *Porta et al., 2018*; *Tsuji et al., 2012*). As for α-synuclein, it has been reported that

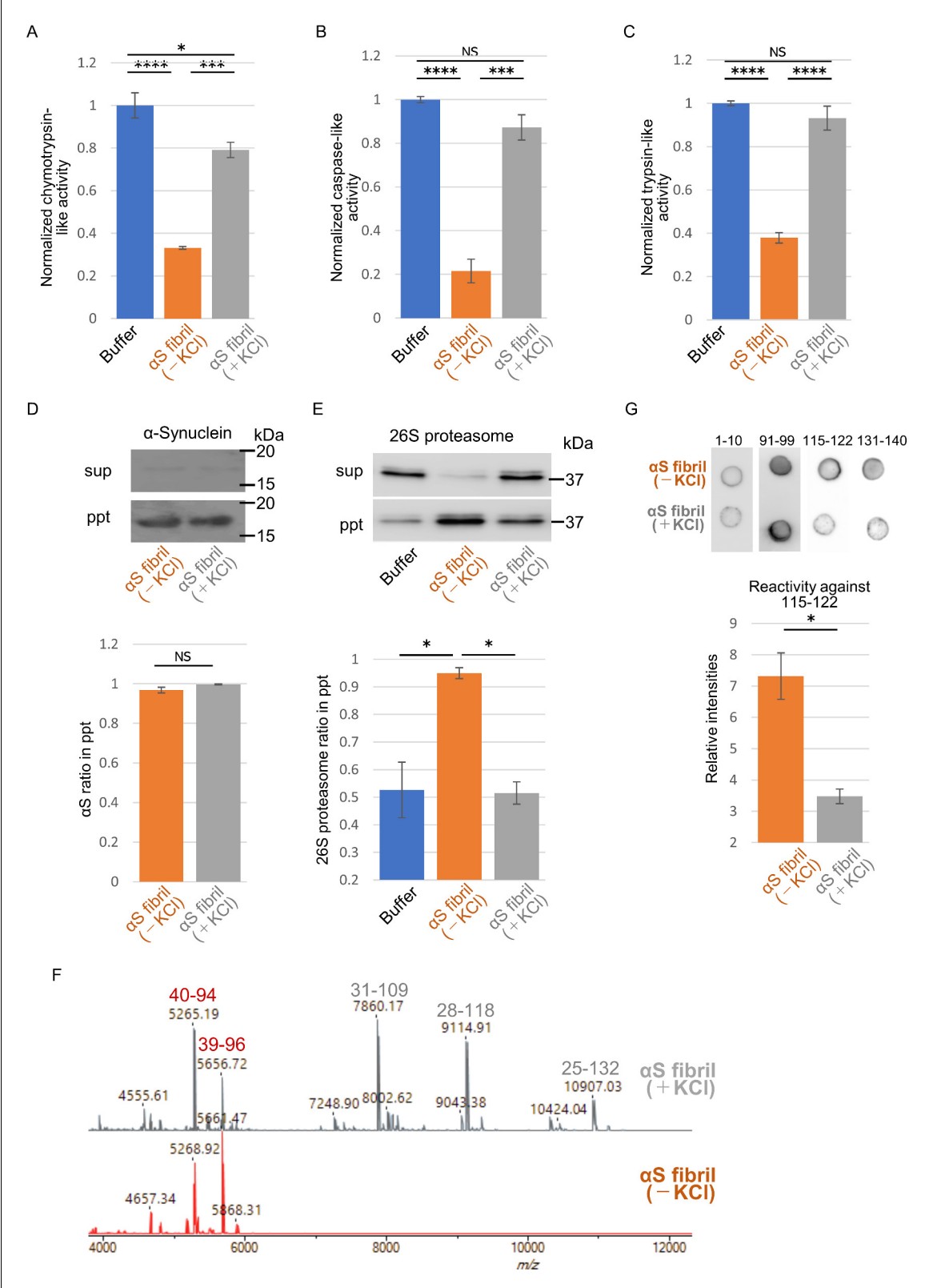

**Figure 4.** Different interaction of α-synuclein strains with 26S proteasome. (**A–C**) Effects of the α-synuclein strains on the 26S proteasome activity. 26S proteasome was purified from budding yeast as shown in *Figure 4—figure supplement 1*. 26S proteasome activities in the presence of the two α-synuclein fibrils were measured. Chymotrypsin like activity was measured by LLVY-MCA hydrolysis(A). Caspase like activity was measured by LLE-MCA hydrolysis (**B**). Trypsin like activity was measured by LRR-MCA hydrolysis (mean ± S.E.M; n = 3). (**D and E**) Co-precipitation of 26S proteasome with the

*Figure 4 continued on next page*

*Figure 4 continued*

α-synuclein strains. 26S proteasome was mixed with the two α-synuclein fibrils and centrifuged. Resulted supernatants (sup) and precipitates (ppt) were analyzed by western blotting against α-synuclein (D, upper) and Rpn11-HA (E, upper). Quantification of the α-synuclein (D, lower) and Rpn11-HA (E, lower) in ppt fractions were shown. (mean ± S.E.M; n = 3) (F) The core regions of the α-synuclein strains. These α-synuclein fibrils were mildly digested by proteinase K and centrifuged. The resulted pellet fractions were denatured by guanidine and analyzed by MALDI-TOF-MS. Peptide peaks identified by mass analysis and the predicted peptide regions corresponding to each peak are shown. (G) Dot blot analysis of the α-synuclein strains. These α-synuclein fibrils were spotted on nitrocellulose membranes and detected by various antibodies against α-synuclein (upper). Reactivity against the antibody raised against 115–122 region of α-synuclein was quantified (mean ± S.E.M; n = 3) (lower). Analysis was performed using one-way ANOVA and Tukey post hoc test. *p<0.05, ***p<0.001, ****p<0.0001 (A, B, C and E). Analysis was performed using student t test. *p<0.05 (D and G).

The online version of this article includes the following source data and figure supplement(s) for figure 4:

**Source data 1.** Quantification for graph in *Figure 4A,B and C*.
**Source data 2.** Quantification for graph in *Figure 4D,E and G*.
**Figure supplement 1.** Purification of 26S proteasome.

aggregates having different structures are formed in vitro (*Bousset et al., 2013*; *Shahnawaz et al., 2020*), and the lesions caused by introduction of aggregates having different structures into cells and rodents are different (*Gribaudo et al., 2019*; *Lau et al., 2020*; *Peelaerts et al., 2015*). However, it has remained unclear what molecular mechanism might lead to the differences in phenotypes induced by different protein aggregates.

In this study, we confirmed that different pathologies were caused by two α-synuclein strains with different structures, and we also found that these α-synuclein strains differ in their ability to inhibit 26S proteasome activity. It has already been reported that protein aggregates that can cause neurodegenerative diseases may interact with proteasome and inhibit its activity (*Bence et al., 2001*; *Thibaudeau et al., 2018*; *Zondler et al., 2017*). However, it is a noteworthy finding in this work that one of the two differently structured fibrils formed from identical monomer under different conditions inhibited proteasome, while the other did not. This clearly raises the possibility that inhibition of proteasome by abnormal α-synuclein plays a role in the pathology. These results provide a possible molecular mechanism to account for the different lesions induced by distinct α-synuclein strains. Considering that PD and MSA have different structures of accumulated α-synuclein (*Klingstedt et al., 2019*; *Lau et al., 2020*; *Prusiner et al., 2015*; *Schweighauser et al., 2020*; *Shahnawaz et al., 2020*; *Woerman et al., 2019*; *Woerman et al., 2015*), it seems highly likely that the structures of α-synuclein fibrils determine the lesions observed in these diseases.

Protein inclusions found in many neurodegenerative diseases are often ubiquitinated, and this is consistent with reports that ubiquitin proteasome systems are impaired in neurons of patients with neurodegenerative diseases (*Bence et al., 2001*). Recently, it was reported that protein aggregates co-aggregate with proteasome at the molecular level (*Guo et al., 2018*). Furthermore, α-synuclein oligomers, Aβ oligomers and polyglutamine inhibit 20S proteasome (*Thibaudeau et al., 2018*). In our α-synuclein fibril formation experiments, almost all the proteins formed insoluble aggregates, and soluble oligomers could not be isolated (*Figure 1—figure supplement 1*). Thus, our results indicate that protein aggregates may also show proteasome-inhibitory activity, and the extent of the inhibitory activity depends on the structure of the aggregates. These facts indicate that pathogenic protein oligomers and aggregates have different effects on cells depending on their structure, and both may be toxic.

Our results suggest that the difference in the structure of the two strains is mainly at the C-terminal region. The C-terminal region (residues 96–140) of α-synuclein is acidic and contains negatively charged residues, including aspartate and glutamate, as well as proline residues. When intermolecular repulsion at the C-terminal region is weakened by changes in ionic strength, exposure of the C-terminal region decreases, and more tightly packed α-synuclein fibrils are formed in the presence of salt. On the other hand, in the absence of salt, intermolecular repulsion at the C-terminal region causes the formation of α-synuclein fibrils in which the C-terminal region is exposed. Only this latter type of α-synuclein fibrils can interact with 26S proteasome complex in vitro, causing inhibition of 26S proteasome activity. If this also occurs in cells, abnormal α-synuclein aggregates, which might be partially degraded by proteasome, would be accumulated (*Figure 6*). More analysis is desired to show that α-synuclein fibrils (-), but not fibrils (+), interact with the mammalian proteasome and inhibit its activity in mouse primary neurons or brains of injected mice. It may also be necessary to

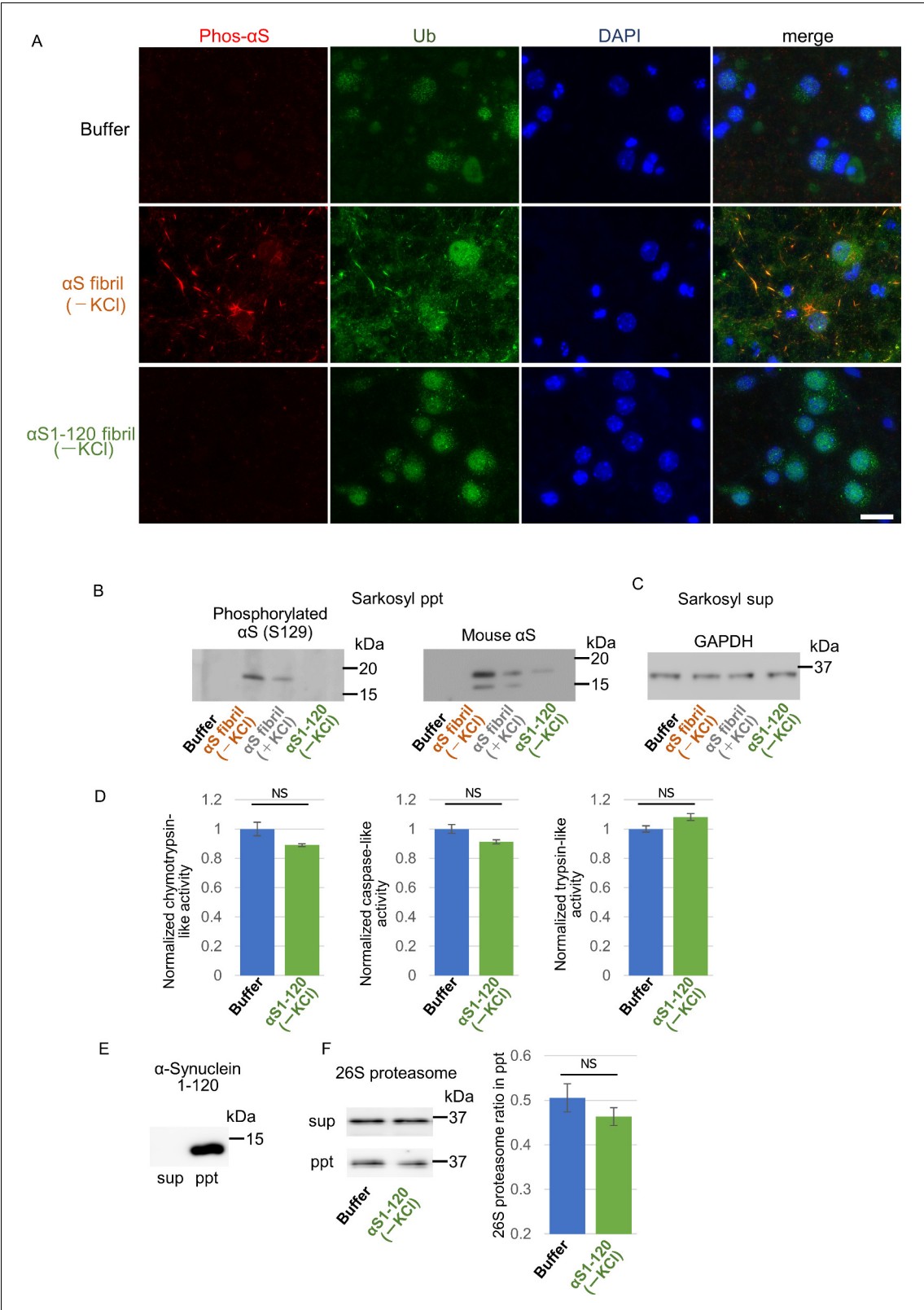

**Figure 5.** Effects of C-terminal truncated α-synuclein fibrils on proteasome in neurons and in vitro. (**A–C**) Seeding activities of C-terminal truncated α-synuclein fibrils in primary mouse cortical neurons. C-terminal truncated α-synuclein fibrils (αSΔC20 (KCl-)) were transduced into primary mouse cortical cells. 14 days after fibril transduction, the accumulation of abnormal phosphorylated α-synuclein and ubiquitinated proteins are detected by immunofluorescence microscopy and western blotting. (**A**) Phosphorylated α-synuclein (Phos-αS), ubiquitin (Ub) and nuclei (DAPI) were stained. Scale

*Figure 5 continued on next page*

*Figure 5 continued*

bar, 20 μm. (**B**) Detection of sarkosyl insoluble phosphorylated α-synuclein (left) and endogenous mouse α-synuclein (right) by western blotting. (**C**) Detection of sarkosyl soluble GAPDH as a loading control. (**D**) Effects of the αSΔC20 (KCl-) fibrils on the 26S proteasome activity. 26S proteasome activities in the presence of the αSΔC20 (KCl-) fibrils were measured. Chymotrypsin like activity (left), caspase like activity (center) and trypsin like activity (right) are shown. (mean ± S.E.M; n = 3). (**E and F**) Co-precipitation of 26S proteasome with the αSΔC20 (KCl-) fibrils. 26S proteasomes were mixed with the αSΔC20 (KCl-) fibrils and centrifuged. Resulted supernatants (sup) and precipitates (ppt) were analyzed by western blotting against α-synuclein (**E**) and Rpn11-HA (**F**, left). Quantification of the Rpn11-HA in ppt fractions were shown (**F**, right). (mean ± S.E.M; n = 3) Analysis was performed using student t test.

The online version of this article includes the following source data for figure 5:

**Source data 1.** Quantification for graph in *Figure 5D and F*.

examine whether protein aggregates co-aggregate with the proteasome in the patient's brain. Consequently, phosphorylated and ubiquitinated α-synuclein aggregates would be accumulated in neurons and then presumably would propagate throughout the brain in a prion-like manner. Our results imply that inhibition of proteasome activity by protein aggregates is critical for prion-like propagation of these protein aggregates. Thus, inhibiting the interaction between aggregates and the proteasome may be a promising therapeutic strategy for neurodegenerative diseases.

Forming fibrils from purified monomeric protein and introducing them into cells or animals might be a useful approach for analyzing the mechanisms of propagation and toxicity of diseases in which protein aggregates accumulate. However, based on our results and previous reports, it seems possible that the results obtained might vary markedly in response to even slight differences in the structure of the introduced fibrils (*Bousset et al., 2013*; *Gribaudo et al., 2019*; *Peelaerts et al., 2015*). Introducing brain extract from a patient with a neurodegenerative disease into cells or animals would also be one approach for identifying the cause of the disease and developing a treatment, but even in patients with the same disease, the results might vary due to slight differences in the structure of the accumulated protein aggregates. Therefore, when conducting such experiments, it is extremely important to analyze the structure of the protein aggregates.

In this study, we found that the degree of interaction with proteasome and the inhibition of proteasome activity vary depending on the strain of α-synuclein aggregates. The relationship between proteasome inhibition and the induction of the pathology supports the hypothesis that prion-like activity of α-synuclein aggregates contributes to disease progression.

# Materials and methods

## Key resources table

| Reagent type (species) or resource | Designation | Source or reference | Identifiers | Additional information |
|---|---|---|---|---|
| Genetic reagent (*Mus musculus*) | C57BL/6J | CLEA, Japan, Inc | RRID:IMSR_JAX:000664 | WT mouse |
| Genetic reagent (*Saccharomyces cerevisiae*) | *RPN11-FLAGx3* | This paper | | |
| Strain, strain background (*Escherichia coli*) | BL21(DE3) | Sigma-Aldrich | Cat. #: 69450 | |
| Recombinant DNA reagent | pRK172-α-synuclein | PMID:8194594 | | |
| Recombinant DNA reagent | pRK172-α-synucleinΔC20 | PMID:30030380 | | |
| Antibody | anti-α-synuclein-pSer129 #64 (mouse monoclonal) | FUJIFILM Wako Chemicals | RRID:AB_2537218 | (1:1000) |

*Continued on next page*

*Continued*

| Reagent type (species) or resource | Designation | Source or reference | Identifiers | Additional information |
|---|---|---|---|---|
| Antibody | anti-α-synuclein 1–10 (rabbit polyclonal) | Cosmo Bio | RRID:AB_2860557 | (1:1000) |
| Antibody | anti-α-synuclein 131–140 (rabbit polyclonal) | Cosmo Bio | RRID:AB_2860558 | (1:1000) |
| Antibody | anti-α-synuclein 91–99 (mouse monoclonal) | BD Bioscience | RRID:AB_398108 | (1:1000) |
| Antibody | anti-α-synuclein-pSer129 (EP1646Y) (rabbit monoclonal) | Abcam | RRID:AB_869971 | (1:1000) |
| Antibody | anti-α-synuclein 115–122 (LB509) (mouse monoclonal) | Santa Cruz Biotechnology | RRID:AB_785898 | (1:200) |
| Antibody | anti-α-synuclein D37A6 (rabbit monoclonal) | Cell Signaling Technology | RRID:AB_1904156 | (1:1000) |
| Antibody | anti-α-synuclein-pSer129 (1175) (rabbit polyclonal) | PMID:23466394 | | (1:1000) |
| Antibody | anti-α-synuclein (Syn205) (mouse monoclonal) | Cell Signaling Technology | RRID:AB_490798 | (1:1000) |
| Antibody | anti-ubiquitin (rabbit polyclonal) | Proteintech | RRID:AB_671515 | (1:1000) |
| Antibody | anti-ubiquitin (rabbit polyclonal) | Dako | RRID:AB_2315524 | (1:10000) |
| Antibody | anti-GAPDH (mouse monoclonal) | Sigma-Aldrich | RRID:AB_2107445 | (1:1000) |
| Antibody | anti-FLAG M2 (mouse monoclonal) | Sigma-Aldrich | RRID:AB_262044 | (1:1000) |
| Antibody | anti-GFAP (D1F4Q) (rabbit monoclonal) | Cell Signaling Technology | RRID:AB_2631098 | (1:200) |
| Antibody | anti-Neurofilament-L (C28E10) (rabbit monoclonal) | Cell Signaling Technology | RRID:AB_823575 | (1:200) |
| Antibody | anti-NeuN (EPR12763) (rabbit monoclonal) | abcam | RRID:AB_2532109 | (1:1000) |
| Antibody | anti-tau (tauC) (rabbit polyclonal) | PMID:26374846 | | (1:200) |
| Antibody | Goat anti-rabbit IgG, Alexa Fluor 488 | Thermo Fisher | RRID:AB_143165 | (1:1000) |
| Antibody | Goat anti-mouse IgG, Alexa Fluor 568 | Thermo Fisher | RRID:AB_2534072 | (1:1000) |
| Antibody | Goat anti-rabbit IgG, HRP | Thermo Fisher | RRID:AB_2536099 | (1:10000) |

*Continued on next page*

*Continued*

| Reagent type (species) or resource | Designation | Source or reference | Identifiers | Additional information |
|---|---|---|---|---|
| Antibody | Goat anti-mouse IgG, HRP | Thermo Fisher | RRID:AB_2536163 | (1:10000) |
| Antibody | Goat anti-rabbit IgG, biotin | Vector laboratories | RRID:AB_2313606 | (1:1000) |
| Software, algorithm | EZR | PMID:23208313 | | |
| Other | hematoxylin | Muto Pure Chemicals | Cat. #: 30002 | |
| Other | DAPI stain | Vector laboratories | Cat #: H-1500 | |

## Expression and purification of recombinant Wild-type and C-terminally truncated human α-synuclein

Full-length and C-terminally truncated α-synuclein encoded in pRK172 plasmids were transformed into *Escherichia coli* BL21 (DE3). Recombinant proteins were purified as described previously (*Nonaka et al., 2010*). Protein concentration was determined by HPLC.

## Preparation of α-synuclein fibrils

α-Synuclein fibrils were prepared as follows. Purified recombinant α-synuclein proteins were dissolved in 30 mM Tris-HCl, pH 7.5, containing 150 mM KCl and 0.1% $NaN_3$, to a final concentration of 6 mg/ml. The samples were incubated at 37°C under rotation at 20 rpm for 7 days. The assembled α-synuclein was sonicated with an ultrasonic homogenizer (VP-5S, TAITEC) in 30 mM Tris-HCl, pH 7.5. For the measurement of turbidity, the resultant α-synuclein assemblies were diluted to 1 mg/ml and the absorbance at 440 nm was measured.

## Transmission electron microscopy

α-Synuclein fibrils were diluted to 15 μM in 30 mM Tris-HCl, pH 7.5, plated on carbon-coated 300-mesh copper grids (Nissin EM), and stained with 2% [v/v] phosphotungstate. Micrographs were recorded on a JEM-1400 electron microscope (JEOL).

## Thioflavin T-binding assay

The degree of fibrillation was measured in terms of Thioflavin T (ThT) fluorescence intensity, which increases when ThT binds to amyloid-like fibrils. The samples (7.5 μM) were incubated with 20 μM ThT in 30 mM Tris-HCl buffer (pH 7.5) for 30 min at 37°C. Fluorometry was performed using a microplate reader (Infinite 200, TECAN, excitation 442 nm, emission 485 nm).

## Congo red binding assay

The binding of Congo red was measured as described previously (*Suzuki et al., 2012*). α-Synuclein monomer and fibrils (37.5 μM) were mixed with Congo red (1 μM) and incubated for 1 hr at 37°C. Absorbance between 400 and 700 nm was measured with a plate reader (Infinite 200, TECAN). The binding of Congo red was calculated as $A_{540}/25296-A_{477}/46306$.

## α-Synuclein aggregation assay

Full-length α-synuclein aggregation experiments were performed using a microplate reader (Infinite 200, TECAN, excitation 442 nm, emission 485 nm) and monitored by measuring ThT fluorescence in the absence or presence of 5% α-syn fibril seeds. All experiments were performed at 37°C, under quiescent conditions in flat-bottomed 96-well black plates (Sumitomo Bakelite) sealed with MicroAmp Optical Adhesive Film (Applied Biosystems). The reaction mixture consisted of PBS containing 1 mg/ml α-synuclein monomer. During experiments under quiescent conditions, ThT fluorescence was read every 2 min.

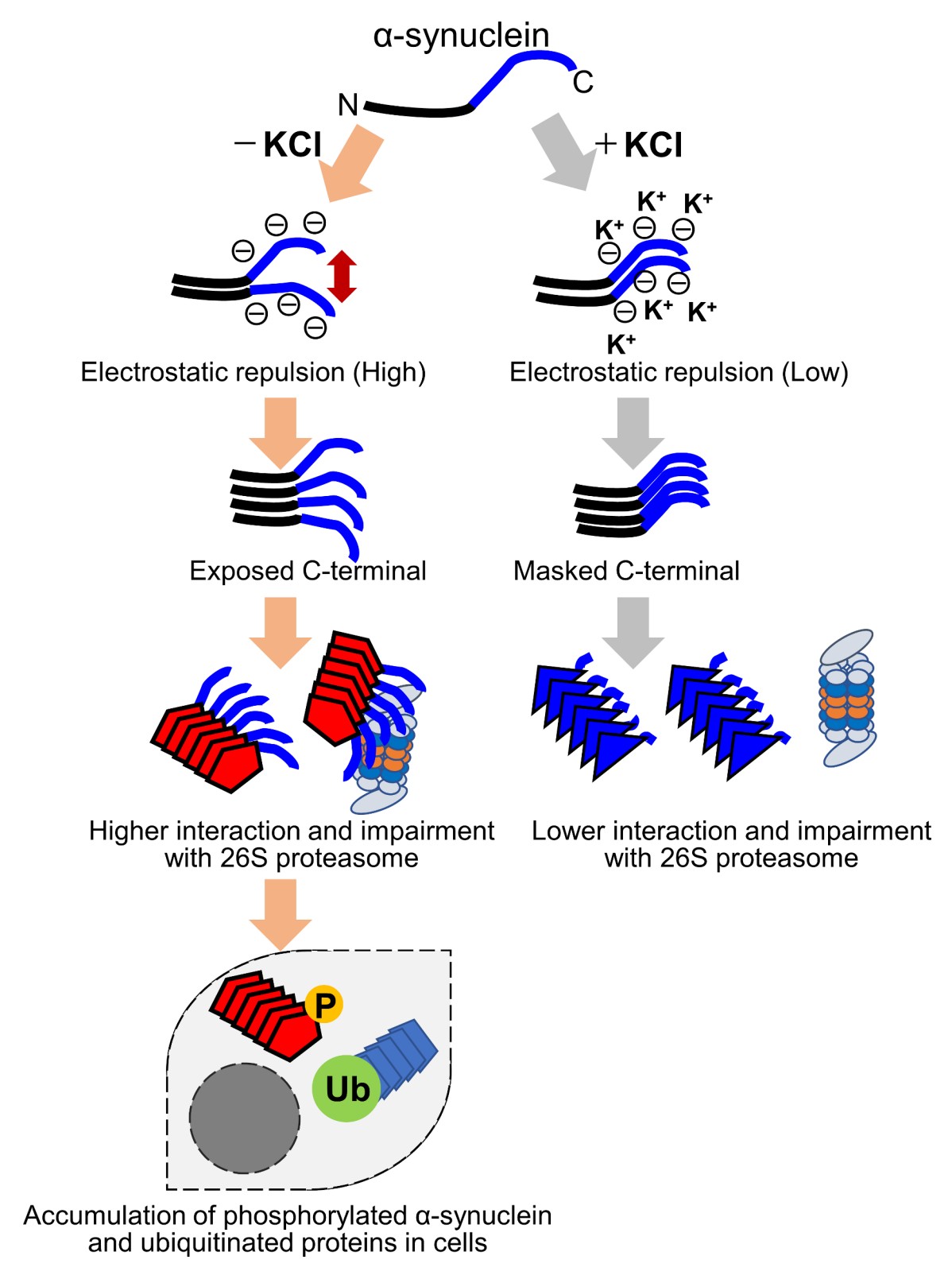

**Figure 6.** Schematic representation of α-synuclein strain formation and strain dependent interaction with 26S proteasome. In the absence of salt, α-synuclein monomers have exposed C-terminal region with high electric repulsion. These form the α-synuclein strain with exposed C-terminal region and this type of the α-synuclein strain can interact with 26S proteasomes and inhibit their activities resulting the accumulation of phosphorylated α-synuclein

*Figure 6 continued on next page*

*Figure 6 continued*

aggregates and ubiquitinated proteins (left). In the presence of salt, α-synuclein monomers have packed C-terminal region with low electric repulsion. These form the α-synuclein strain with masked C-terminal region and this type of the α-synuclein strain can not interact with 26S proteasomes.

### Primary-cultured cells and introduction of α-synuclein proteins into cells

Dissociated cultures of embryonic (E15) mouse cortical cells were prepared from pregnant C57BL/6 mice using Neuron Dissociation Solutions (FUJI FILM Wako) according to the manufacturer's protocol. Briefly, dissected brain was digested with enzyme solution for 30 min at 37°C, then centrifuged. Dispersion solution was added and tissues were suspended, then isolation solution was added. Cells were collected by centrifugation, resuspended and plated on poly-L-lysine-coated cover glass or plates. Cells were maintained at 37°C in 5% $CO_2$ in Neurobasal Medium (Gibco) supplemented with 1x B27 and 1x Glutamax. Cells were cultured for 7 days in vitro (DIV) in 6-well plates, and then treated with sonicated α-synuclein fibrils diluted in culture medium. Cells were collected or fixed at 14 days post treatment (21 DIV).

### Purification of 26S Proteasome from Budding Yeast

Yeast 26S proteasome was purified as described previously (*Saeki et al., 2005*). Briefly, the yeast strain (BY4741, *Rpn11-FLAGx3::KanMX*) was cultured in YPD for 2 days, harvested, washed and stocked at −80°C. Buffer A'' (50 mM Tris-HCl, pH 7.5, 100 mM NaCl, 10% glycerol, 4 mM ATP, 10 mM $MgCl_2$) and glass beads were added and the cells were lysed in a Beads Shocker (Yasui Kikai). Cell debris was removed by centrifugation and then anti-FLAG M2 antibody-conjugated agarose beads (Sigma) were added. The mixture was incubated for 2 hr at 4°C. Agarose beads were washed with Buffer A'' and Buffer A'' containing 0.1% Triton, and proteasome was eluted by adding 3x FLAG peptide (400 µg/ml) (Sigma) in Buffer A''.

### 26S proteasome activity assays

Inhibition of α-synuclein fibrils on proteasome activity was measured using fluorogenic peptides in 96-well black flat-bottomed plates. 26S proteasome (10 µg/ml) was added to α-synuclein fibrils (35 µM, calculated based on the monomer protein concentration), and the mixture was incubated in buffer A'' containing 100 µM fluorogenic substrate (suc-LLVY-mca, Z-LLE-mca, Peptide Institute) for 60 min at 37°C or 50 µM fluorogenic substrate (boc-LRR-mca, Peptide Institute) for 10 min at 37°C. Fluorescence was measured before and after incubation (Infinite 200, TECAN, excitation 360 nm, emission 440 nm). The rate of increase in fluorescence intensity is regarded as representing proteasome activity.

### Binding assay of proteasome with α-synuclein fibrils

α-Synuclein fibrils were mixed with purified 26S proteasome in Buffer A'', and the mixture was centrifuged at 21,500 x g for 20 min. The supernatant and pellet fractions were analyzed by the western blotting using the appropriate antibodies (RRID:AB_262044, RRID:AB_2860557, RRID:AB_2860558).

### Immunocytochemistry

Introduction of α-synuclein fibrils were conducted as described above, using mouse primary cultured cells grown on coverslips. At 2 weeks after introduction of fibrils, the cells were fixed with 4% paraformaldehyde and treated with the primary antibodies (RRID:AB_2537218, RRID:AB_671515, RRID:AB_2631098, RRID:AB_823575, RRID:AB_2532109, tauC). After incubation overnight, the cells were washed and treated with secondary antibodies conjugated with Alexa Fluor (RRID:AB_143165, RRID:AB_2534072) for 1 hr. The cells were mounted with DAPI to counterstain nuclear DNA and analyzed with a BZ-X710 (Keyence) and BZ-X analyzer (Keyence).

### α-Synuclein inoculation into mice and immunohistochemistry of mouse brains

Ten-week-old, male C57BL/6J mice were purchased from CLEA Japan, Inc All experimental protocols were performed according to the recommendations of the Animal Care and Use Committee of Tokyo Metropolitan Institute of Medical Science. α-Synuclein samples (150 µM, 5 µl) were injected

into the striatum (anterior-posterior, 0.2 mm; medial-lateral, −2.0 mm; dorsal-ventral, 2.6 mm). Inoculation into mouse brain was performed as described previously (*Masuda-Suzukake et al., 2013*). Non-injected wild- type mice were used as negative controls.

One month after inoculation, mice were deeply anesthetized with isoflurane (Pfizer) and sacrificed, and the brain was perfused with 0.1 M phosphate buffer. Sections were fixed in 4% paraformaldehyde and preserved in 20% sucrose in 0.01 M phosphate buffered saline, pH 7.4. Sections were cut serially on a freezing microtome at 30 μm thickness. Sections were then mounted on glass slides. Sections were incubated with 1% $H_2O_2$ for 30 min to eliminate endogenous peroxidase activity and were treated with 100% formic acid (Wako) for 10 min for antigen retrieval and washed under running tap water. Immunohistochemistry with polyclonal antibody 1175 (1:1,000) directed against α-synuclein phosphorylated at Ser129 or anti-ubiquitin (RRID:AB_2315524) were performed as described previously (*Masuda-Suzukake et al., 2013*). Antibody labeling was performed by incubation with biotinylated goat anti-rabbit IgG (RRID:AB_2313606) for 3 hr. The antibody labeling was visualized by incubation with avidin-biotinylated horseradish peroxidase complex (ABC Elite, Vector Laboratories, 1:1,000) for 3 hr, followed by incubation with a solution containing 0.01% 3,3'-diaminobenzidine, 0.05 M imidazole and 0.00015% $H_2O_2$ in 0.05 M Tris-HCl buffer, pH 7.6. Counter nuclear staining was performed with hematoxylin (Muto Pure Chemicals). The sections were then rinsed with distilled water, treated with xylene, and coverslipped with Entellan (Merck). Images were analyzed with a BZ-X710 (Keyence) and BZ-X analyzer (Keyence). We examined three mice for each sample and each mouse yielded highly similar results. We examined at least 6 slices for each mouse and confirmed the similar results. Animal experiments were done by the experimenter who is blind.

## Sedimentation analysis and western blotting

Cells were harvested, collected by centrifugation (2,000 × g, 5 min) and washed with PBS. The cellular proteins were extracted by sonication in 200 μl of buffer A68 (10 mM Tris-HCl, pH 7.5, 1 mM EGTA, 10% sucrose, 0.8 M NaCl containing sarkosyl (final 1%, w/v) and protease inhibitor (Roche). After ultracentrifugation at 135,000 × g for 20 min at 25˚C, the supernatant was collected as sarkosyl-soluble fraction, and the protein concentration was determined by Bradford assay. The pellet was solubilized in 50 μl of SDS-sample buffer. Both sarkosyl-soluble and insoluble fractions were analyzed by immunoblotting with the appropriate antibodies (RRID:AB_869971, RRID:AB_1904156, RRID:AB_671515, RRID:AB_2107445).

## Dot blot analysis

α-synuclein fibrils (50 ng) were spotted on nitrocellulose membranes and detected by the appropriate antibodies against α-synuclein (RRID:AB_2860557, RRID:AB_2860558, RRID:AB_398108, RRID:AB_785898).

## Fibril uptake assay

Mouse cortical cells were prepared and treated with the α-synuclein fibrils as described above. A day after treatment, cells were washed with PBS and treated with 0.25% trypsin at 37˚C for 10 min, then cells were collected. Sedimentation analysis and western blotting were performed as described above using the appropriate antibodies (RRID:AB_490798, RRID:AB_2107445).

## Statistical analysis

Student's *t*-test was performed when comparing 2 groups. One-way ANOVA and Tukey's post hoc test were performed with EZR when comparing 3 groups (*Kanda, 2013*). P values below 0.05 were considered to be statistically significant.

## Acknowledgements

We would like to thank all members of the laboratory for helpful discussion.

## Additional information

### Funding

| Funder | Grant reference number | Author |
| --- | --- | --- |
| Japan Society for the Promotion of Science | 16K21650 | Genjiro Suzuki |
| Ichiro Kanehara Foundation for the Promotion of Medical Sciences and Medical Care | | Genjiro Suzuki |
| Kato Memorial Bioscience Foundation | | Genjiro Suzuki |
| Ministry of Education, Culture, Sports, Science, and Technology | 26117005 | Masato Hasegawa |
| Core Research for Evolutional Science and Technology | JPMJCR18H3 | Masato Hasegawa |
| Japan Agency for Medical Research and Development | JP18dm0207019 | Masato Hasegawa |

The funders had no role in study design, data collection and interpretation, or the decision to submit the work for publication.

### Author contributions

Genjiro Suzuki, Conceptualization, Resources, Data curation, Formal analysis, Funding acquisition, Validation, Investigation, Methodology, Writing - original draft, Writing - review and editing; Sei Imura, Conceptualization, Investigation, Methodology; Masato Hosokawa, Ryu Katsumata, Investigation, Methodology; Takashi Nonaka, Conceptualization, Writing - review and editing; Shin-Ichi Hisanaga, Conceptualization, Supervision; Yasushi Saeki, Conceptualization, Resources, Methodology; Masato Hasegawa, Conceptualization, Resources, Supervision, Funding acquisition, Project administration, Writing - review and editing

### Author ORCIDs

Genjiro Suzuki (iD) https://orcid.org/0000-0002-1400-4139
Takashi Nonaka (iD) http://orcid.org/0000-0002-0830-9403

### Ethics

Animal experimentation: All experimental protocols were performed according to the recommendations of the Animal Care and Use Committee of Tokyo Metropolitan Institute of Medical Science (#18040, #19042, #20-035).

### Decision letter and Author response

Decision letter https://doi.org/10.7554/eLife.56825.sa1
Author response https://doi.org/10.7554/eLife.56825.sa2

## Additional files

### Supplementary files

• Transparent reporting form

### Data availability

All data generated or analysed during this study are included in the manuscript and supporting files. Source data files have been provided for Figures 1, 3, 4 and 5.

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
