## [Decision Letter]

**Acceptance summary:**

You revealed that two types of α-synuclein fibrils, which are generated from identical monomers but have different structural properties, show a drastic difference in their abilities to induce formation of aggregates containing α-synuclein and ubiquitin in cultured neurons and mice brain, and also elucidated the underlying mechanism. Thus, this study provides significant insights into how distinct α-synuclein strains lead to different pathologies, an important subject in basic medical science.

**Decision letter after peer review:**

Thank you for submitting your article "α-Synuclein strains that cause distinct pathologies differentially inhibit proteasome" for consideration by *eLife*. Your article has been reviewed by three peer reviewers, one of whom is a member of our Board of Reviewing Editors, and the evaluation has been overseen by David Ron as the Senior Editor.

The reviewers have discussed the reviews with one another and the Reviewing Editor has drafted this decision to help you prepare a revised submission.

Summary:

In this study, Suzuki et al. generated two types of α-synuclein fibrils from identical monomers in vitro and investigated their seeding and propagation ability in mice and primary-cultured neurons.

One of the strains induced phosphorylated α-synuclein- and ubiquitin-positive aggregates in cultured neurons and mice brain, whereas the other strain, which formed fibrils more efficiently in vitro, did not. The authors further obtained results suggesting that α-synuclein in the former fibrils adopts a more flexible conformation in its C-terminal region, which interacts with the proteasome and thereby inhibits its activity. Overall, we find this study provides significant insights into how distinct α-synuclein strains lead to different pathologies, an important subject in basic medical science, but some critical issues remain to be addressed.

Major revisions:

1) The authors should include a control in their in vivo study and show it in the immunohistochemistry analysis. The different panels should be labelled with the treatments used. Furthermore, it should be specified how many mice have been used for this experiment. As there is no quantification data presented, it would be nice to mention how many different images were analyzed qualitatively from each condition.

2) The authors should improve the quality of the IF images in Figure 3A, B. Moreover, they should include the staining of the nuclei in Figure 3A. The nuclei appear to have different size in diameter, the authors should perform a staining for neuronal and astrocyte markers. Furthermore, they should specify what kind of primary neurons they used.

3) Figure 3D, E: in the WB images, the authors have to show the loading control (e.g., GAPDH).

4) Figure 4—figure supplement 2A: the authors should improve the quality of the IF images and include the IF related to the α-synuclein fibrils (-) not truncated and the control without fibrils.

5) Figure 4—figure supplement 2B, C: in the WB images, the authors have to include the loading control (e.g., GAPDH).

6) "Considering all these results, we can conclude that the C-terminal region of α-synuclein is exposed only in α-synuclein fibrils (-) and this region interacts with 26S proteasome and impairs its activity." To assess these conclusions, the authors should perform in vitro experiments shown in Figure 4 for ΔC20 fibrils (-); whether these fibrils indeed do not interact with the proteasome and inhibit its activity.

7) The authors should perform co-immunoprecipitation analysis using primary neurons treated with α-synuclein fibrils (-) to confirm that α-synuclein fibrils (-), but not fibrils (+), interact with the mammalian proteasome.

8) The authors should confirm that the two α-synuclein fibrils were taken up into brain/neuronal cells with similar efficiency. Because different turbidity in these fibril samples suggests that their fibril (particle) sizes are not the same, the authors should carefully consider the possibility that fibrils (+), which might be larger than fibrils (-), were not efficiently introduced into the cells.

---

## [Author Response]

Major revisions:1) The authors should include a control in their in vivo study and show it in the immunohistochemistry analysis. The different panels should be labelled with the treatments used. Furthermore, it should be specified how many mice have been used for this experiment. As there is no quantification data presented, it would be nice to mention how many different images were analyzed qualitatively from each condition.

We appreciate the reviewer’s helpful comment. We added negative control images and labels with the treatments in Figure 2. We added the information about how many mice have been used in Figure 2—figure supplement 2 and subsection “Formation of

Phosphorylated α-Synuclein Pathology Induced by the Two α-Synuclein Strains in Cultured Neurons”. We also added the information about how many different slices used in subsection “α-Synuclein Inoculation into Mice and Immunohistochemistry of Mouse Brains”.

2) The authors should improve the quality of the IF images in Figure 3A, B. Moreover, they should include the staining of the nuclei in Figure 3A. The nuclei appear to have different size in diameter, the authors should perform a staining for neuronal and astrocyte markers. Furthermore, they should specify what kind of primary neurons they used.

We united Figure 3A with 3B, improved the quality of images and included the staining of the nuclei in revised Figure 3A. We added the double staining images for neuronal and astrocyte markers in Figure 3—figure supplement 1. As the reviewer’s indication, we used mouse cortical primary cells, not mouse primary neurons, in this study. We changed the description about mouse primary neurons in the revised text.

3) Figure 3D, E: in the WB images, the authors have to show the loading control (e.g., GAPDH).

The loading control for Figure 3D and E is the same that used in Figure 3C because the samples used in these figures are the same. We apologize for our confusing presentation in Figure 3C-E. In the revised Figure 3, we showed sarkosyl insoluble fractions in Figure 3B and sarlosyl soluble fractions in Figure 3C as the loading control. We appreciate the reviewer’s helpful comment.

4) Figure 4—figure supplement 2A: the authors should improve the quality of the IF images and include the IF related to the α-synuclein fibrils (-) not truncated and the control without fibrils.

We improved the quality of IF images and added the IF imaged of α-synuclein fibrils (-) not truncated and the buffer control in new Figure 5A.

5) Figure 4—figure supplement 2B, C: in the WB images, the authors have to include the loading control (e.g., GAPDH).

We added the loading control in new Figure 5C.

6) "Considering all these results, we can conclude that the C-terminal region of α-synuclein is exposed only in α-synuclein fibrils (-) and this region interacts with 26S proteasome and impairs its activity." To assess these conclusions, the authors should perform in vitro experiments shown in Figure 4 for ΔC20 fibrils (-); whether these fibrils indeed do not interact with the proteasome and inhibit its activity.

We added in vitro experiments about the effect of the ΔC20 fibrils (-) on the proteasome activity in Figure 5D and about the interaction between ΔC20 fibrils (-) and the proteasome in Figure 5E and F.

7) The authors should perform co-immunoprecipitation analysis using primary neurons treated with α-synuclein fibrils (-) to confirm that α-synuclein fibrils (-), but not fibrils (+), interact with the mammalian proteasome.

We thank the reviewer’s constructive comment. We tried to detect direct interactions between the proteasome complex and α-synuclein fibrils (-), but not α-synuclein fibrils (+) in mouse primary cortical cells, but we could not detect such interactions by co-immunoprecipitation. In our opinion, it is because that both the proteasome complex and α-synuclein are abundant in the cells, however, the inhibitory complex between the proteasome and α-synuclein fibrils must be little compared with the total amounts of these proteins. We will continue to detect interactions between the proteasome complex and α-synuclein fibrils (-) by co-immunoprecipitation or other methods and we will report the results in a preprint on bioRxiv or a Research Advance in *eLife*. We added the discussion about the importance of these experiments in Discussion paragraph four.

8) The authors should confirm that the two α-synuclein fibrils were taken up into brain/neuronal cells with similar efficiency. Because different turbidity in these fibril samples suggests that their fibril (particle) sizes are not the same, the authors should carefully consider the possibility that fibrils (+), which might be larger than fibrils (-), were not efficiently introduced into the cells.

We appreciate the reviewer’s helpful comment. We investigated the uptake of these fibrils into primary cells by western blotting of the cells corrected a day after fibril treatment. Briefly, cells were treated with the two α-synuclein fibrils. A day after treatment, we treated cells with trypsin for digestion of extracellular fibrils, collected them and examined the α-synuclein fibrils taken up into the cells by western blotting of sarkosyl insoluble fractions. Surprisingly, we found α-synuclein fibrils (-), which induced much α-synuclein accumulation, were taken up into the cells lesser than α-synuclein fibrils (+), indicating that α-synuclein fibrils (-) efficiently caused the accumulation of α-synuclein and ubiquitinated proteins in spite of their inefficient uptake into the cells. We added the result and the method in Figure 3—figure supplement 2, subsection “Formation of Phosphorylated α-Synuclein Pathology Induced by the Two α-Synuclein Strains in Cultured Neurons” paragraph two and subsection “Fibril uptake assay”.